# GRADIENT ORIGIN NETWORKS

**Sam Bond-Taylor*** & **Chris G. Willcocks***
Department of Computer Science
Durham University
{samuel.e.bond-taylor,christopher.g.willcocks}@durham.ac.uk

## ABSTRACT

This paper proposes a new type of generative model that is able to quickly learn a latent representation without an encoder. This is achieved using empirical Bayes to calculate the expectation of the posterior, which is implemented by initialising a latent vector with zeros, then using the gradient of the log-likelihood of the data with respect to this zero vector as new latent points. The approach has similar characteristics to autoencoders, but with a simpler architecture, and is demonstrated in a variational autoencoder equivalent that permits sampling. This also allows implicit representation networks to learn a space of implicit functions without requiring a hypernetwork, retaining their representation advantages across datasets. The experiments show that the proposed method converges faster, with significantly lower reconstruction error than autoencoders, while requiring half the parameters.

## 1 INTRODUCTION

Observable data in nature has some parameters which are known, such as local coordinates, but also some unknown parameters such as how the data is related to other examples. Generative models, which learn a distribution over observables, are central to our understanding of patterns in nature and allow for efficient query of new unseen examples. Recently, deep generative models have received interest due to their ability to capture a broad set of features when modelling data distributions. As such, they offer direct applications such as synthesising high fidelity images (Karras et al., 2020), super-resolution (Dai et al., 2019), speech synthesis (Li et al., 2019), and drug discovery (Segler et al., 2018), as well as benefits for downstream tasks like semi-supervised learning (Chen et al., 2020).

A number of methods have been proposed such as Variational Autoencoders (VAEs, Figure 1a), which learn to encode the data to a latent space that follows a normal distribution permitting sampling (Kingma & Welling, 2014). Generative Adversarial Networks (GANs) have two competing networks, one which generates data and another which discriminates from implausible results (Goodfellow et al., 2014). Variational approaches that approximate the posterior using gradient descent (Lipton & Tripathi, 2017) and short run MCMC (Nijkamp et al., 2020) respectively have been proposed, but to obtain a latent vector for a sample, they require iterative gradient updates. Autoregressive Models (Van Den Oord et al., 2016) decompose the data distribution as the product of conditional distributions and Normalizing Flows (Rezende & Mohamed, 2015) chain together invertible functions; both methods allow exact likelihood inference. Energy-Based Models (EBMs) map data points to

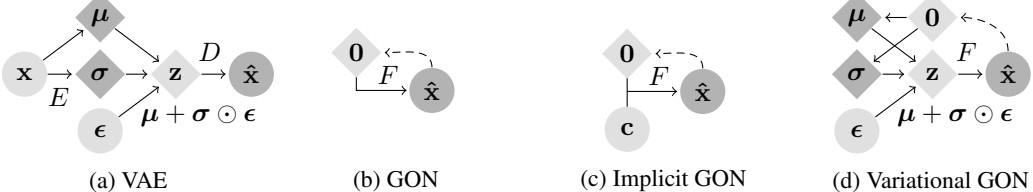

|  |  |  |  |
|---|---|---|---|
| (a) VAE | (b) GON | (c) Implicit GON | (d) Variational GON |

Figure 1: Gradient Origin Networks (GONs; b) use gradients (dashed lines) as encodings thus only a single network $F$ is required, which can be an implicit representation network (c). Unlike VAEs (a) which use two networks, $E$ and $D$, variational GONs (d) permit sampling with only one network.

---

*Authors contributed equally.

energy values proportional to likelihood thereby permitting sampling through the use of Monte Carlo Markov Chains (Du & Mordatch, 2019). In general to support encoding, these approaches require separate encoding networks, are limited to invertible functions, or require multiple sampling steps.

Implicit representation learning (Park et al., 2019; Tancik et al., 2020), where a network is trained on data parameterised continuously rather than in discrete grid form, has seen a surge of interest due to the small number of parameters, speed of convergence, and ability to model fine details. In particular, sinusoidal representation networks (SIRENs) (Sitzmann et al., 2020b) achieve impressive results, modelling many signals with high precision, thanks to their use of periodic activations paired with carefully initialised MLPs. So far, however, these models have been limited to modelling single data samples, or use an additional hypernetwork or meta learning (Sitzmann et al., 2020a) to estimate the weights of a simple implicit model, adding significant complexity.

This paper proposes Gradient Origin Networks (GONs), a new type of generative model (Figure 1b) that do not require encoders or hypernetworks. This is achieved by initialising latent points at the origin, then using the gradient of the log-likelihood of the data with respect to these points as the latent space. At inference, latent vectors can be obtained in a single step without requiring iteration. GONs are shown to have similar characteristics to convolutional autoencoders and variational autoencoders using approximately half the parameters, and can be applied to implicit representation networks (such as SIRENs) allowing a space of implicit functions to be learned with a simpler overall architecture.

## 2 PRELIMINARIES

We first introduce some background context that will be used to derive our proposed approach.

### 2.1 EMPIRICAL BAYES

The concept of empirical Bayes (Robbins, 1956; Saremi & Hyvarinen, 2019), for a random variable $\mathbf{z} \sim p_\mathbf{z}$ and particular observation $\mathbf{z}_0 \sim p_{\mathbf{z}_0}$, provides an estimator of $\mathbf{z}$ expressed purely in terms of $p(\mathbf{z}_0)$ that minimises the expected squared error. This estimator can be written as a conditional mean:

$$\hat{\mathbf{z}}(\mathbf{z}_0) = \int \mathbf{z} p(\mathbf{z}|\mathbf{z}_0) d\mathbf{z} = \int \mathbf{z} \frac{p(\mathbf{z}, \mathbf{z}_0)}{p(\mathbf{z}_0)} d\mathbf{z}. \tag{1}$$

Of particular relevance is the case where $\mathbf{z}_0$ is a noisy observation of $\mathbf{z}$ with covariance $\mathbf{\Sigma}$. In this case $p(\mathbf{z}_0)$ can be represented by marginalising out $\mathbf{z}$:

$$p(\mathbf{z}_0) = \int \frac{1}{(2\pi)^{d/2} |\det(\mathbf{\Sigma})|^{1/2}} \exp\Big( -(\mathbf{z}_0 - \mathbf{z})^T \mathbf{\Sigma}^{-1} (\mathbf{z}_0 - \mathbf{z})/2 \Big) p(\mathbf{z}) d\mathbf{z}. \tag{2}$$

Differentiating this with respect to $\mathbf{z}_0$ and multiplying both sides by $\mathbf{\Sigma}$ gives:

$$\mathbf{\Sigma} \nabla_{\mathbf{z}_0} p(\mathbf{z}_0) = \int (\mathbf{z} - \mathbf{z}_0) p(\mathbf{z}, \mathbf{z}_0) d\mathbf{z} = \int \mathbf{z} p(\mathbf{z}, \mathbf{z}_0) d\mathbf{z} - \mathbf{z}_0 p(\mathbf{z}_0). \tag{3}$$

After dividing through by $p(\mathbf{z}_0)$ and combining with Equation 1 we obtain a closed form estimator of $\mathbf{z}$ (Miyasawa, 1961) written in terms of the score function $\nabla \log p(\mathbf{z}_0)$ (Hyvärinen, 2005):

$$\hat{\mathbf{z}}(\mathbf{z}_0) = \mathbf{z}_0 + \mathbf{\Sigma} \nabla_{\mathbf{z}_0} \log p(\mathbf{z}_0). \tag{4}$$

This optimal procedure is achieved in what can be interpreted as a single gradient descent step, with no knowledge of the prior $p(\mathbf{z})$. By rearranging Equation 4, a definition of $\nabla \log p(\mathbf{z}_0)$ can be derived; this can be used to train models that approximate the score function (Song & Ermon, 2019).

### 2.2 VARIATIONAL AUTOENCODERS

Variational Autoencoders (VAEs; Kingma & Welling 2014) are a probabilistic take on standard autoencoders that permit sampling. A latent-based generative model $p_{\boldsymbol{\theta}}(\mathbf{x}|\mathbf{z})$ is defined with a normally distributed prior over the latent variables, $p_{\boldsymbol{\theta}}(\mathbf{z}) = \mathcal{N}(\mathbf{z}; \mathbf{0}, \boldsymbol{I}_d)$. $p_{\boldsymbol{\theta}}(\mathbf{x}|\mathbf{z})$ is typically parameterised as a Bernoulli, Gaussian, multinomial distribution, or mixture of logits. In this case, the true posterior $p_{\boldsymbol{\theta}}(\mathbf{z}|\mathbf{x})$ is intractable, so a secondary encoding network $q_{\boldsymbol{\phi}}(\mathbf{z}|\mathbf{x})$ is used to approximate the true posterior; the pair of networks thus resembles a traditional autoencoder. This allows VAEs to approximate $p_{\boldsymbol{\theta}}(\mathbf{x})$ by maximising the evidence lower bound (ELBO), defined as:

$$\log p_{\boldsymbol{\theta}}(\mathbf{x}) \geq \mathcal{L}^{\text{VAE}} = -D_{\text{KL}}(\mathcal{N}(q_{\boldsymbol{\phi}}(\mathbf{z}|\mathbf{x}))||\mathcal{N}(\mathbf{0}, \boldsymbol{I}_d)) + \mathbb{E}_{q_{\boldsymbol{\phi}}(\mathbf{z}|\mathbf{x})}[\log p_{\boldsymbol{\theta}}(\mathbf{x}|\mathbf{z})]. \tag{5}$$

To optimise this lower bound with respect to $\boldsymbol{\theta}$ and $\boldsymbol{\phi}$, gradients must be backpropagated through the stochastic process of generating samples from $\mathbf{z}' \sim q_{\boldsymbol{\phi}}(\mathbf{z}|\mathbf{x})$. This is permitted by reparameterising $\mathbf{z}$ using the differentiable function $\mathbf{z}' = \mu(\mathbf{z}) + \sigma(\mathbf{z}) \odot \boldsymbol{\epsilon}$, where $\boldsymbol{\epsilon} \sim \mathcal{N}(\mathbf{0}, \boldsymbol{I}_d)$ and $\mu(\mathbf{z})$ and $\sigma(\mathbf{z})^2$ are the mean and variance respectively of a multivariate Gaussian distribution with diagonal covariance.

## 3 METHOD

Consider some dataset $\mathbf{x} \sim p_d$ of continuous or discrete signals $\mathbf{x} \in \mathbb{R}^m$, it is typical to assume that the data can be represented by low dimensional latent variables $\mathbf{z} \in \mathbb{R}^k$, which can be used by a generative neural network to reconstruct the data. These variables are often estimated through the use of a secondary encoding network that is trained concurrently with the generative network. An encoding network adds additional complexity (and parameters) to the model, it can be difficult to balance capacities of the two networks, and for complex hierarchical generative models designing a suitable architecture can be difficult. This has led some to instead approximate latent variables by performing gradient descent on the generative network (Bojanowski et al., 2018; Nijkamp et al., 2020). While this addresses the aforementioned problems, it significantly increases the run time of the inference process, introduces additional hyperparameters to tune, and convergence is not guaranteed.

### 3.1 GRADIENT ORIGIN NETWORKS

We propose a generative model that consists only of a decoding network, using empirical Bayes to approximate the posterior in a single step. That is, for some data point $\mathbf{x}$ and latent variable $\mathbf{z} \sim p_{\mathbf{z}}$, we wish to find an approximation of $p(\mathbf{z}|\mathbf{x})$. Given some noisy observation $\mathbf{z}_0 = \mathbf{z} + \mathcal{N}(\mathbf{0}, \boldsymbol{I}_d)$ of $\mathbf{z}$ then empirical Bayes can be applied to approximate $\mathbf{z}$. Specifically, since we wish to approximate $\mathbf{z}$ conditioned on $\mathbf{x}$, we instead calculate $\hat{\mathbf{z}}_{\mathbf{x}}$, the least squares estimate of $p(\mathbf{z}|\mathbf{x})$ (proof in Appendix A):

$$\hat{\mathbf{z}}_{\mathbf{x}}(\mathbf{z}_0) = \mathbf{z}_0 + \nabla_{\mathbf{z}_0} \log p(\mathbf{z}_0|\mathbf{x}). \tag{6}$$

Using Bayes' rule, $\log p(\mathbf{z}_0|\mathbf{x})$ can be written as $\log p(\mathbf{z}_0|\mathbf{x}) = \log p(\mathbf{x}|\mathbf{z}_0) + \log p(\mathbf{z}_0) - \log p(\mathbf{x})$. Since $\log p(\mathbf{x})$ is a normalising constant that does not affect the gradient, we can rewrite Equation 6 in terms only of the decoding network and $p(\mathbf{z}_0)$:

$$\hat{\mathbf{z}}_{\mathbf{x}}(\mathbf{z}_0) = \mathbf{z}_0 + \nabla_{\mathbf{z}_0} \Big( \log p(\mathbf{x}|\mathbf{z}_0) + \log p(\mathbf{z}_0) \Big). \tag{7}$$

It still remains, however, how to construct a noisy estimate of $\mathbf{z}_0$ with no knowledge of $\mathbf{z}$. If we assume $\mathbf{z}$ follows a known distribution, then it is possible to develop reasonable estimates. For instance, if we assume $p(\mathbf{z}) = \mathcal{N}(\mathbf{z}; \mathbf{0}, \boldsymbol{I}_d)$ then we could sample from $p(\mathbf{z}_0) = \mathcal{N}(\mathbf{z}_0; \mathbf{0}, 2\boldsymbol{I}_d)$ however this could be far from the true distribution of $p(\mathbf{z}_0|\mathbf{z}) = \mathcal{N}(\mathbf{z}_0; \mathbf{z}, \boldsymbol{I}_d)$. Instead we propose initialising $\mathbf{z}_0$ at the origin since this is the distribution's mean. Initialising at a constant position decreases the input variation and thus simplifies the optimisation procedure. Naturally, how $p(\mathbf{x}|\mathbf{z})$ is modelled affects $\hat{\mathbf{z}}_{\mathbf{x}}$. While mean-field models result in $\hat{\mathbf{z}}_{\mathbf{x}}$ that are linear functions of $\mathbf{x}$, conditional autoregressive models, for instance, result in non-linear $\hat{\mathbf{z}}_{\mathbf{x}}$; multiple gradient steps also induce non-linearity, however, we show that a single step works well on high dimensional data suggesting that linear encoders, which normally do not scale to high dimensional data are effective in this case.

### 3.2 AUTOENCODING WITH GONS

Before exploring GONs as generative models, we discuss the case where the prior $p(\mathbf{z})$ is unknown; such a model is referred to as an autoencoder. As such, the distribution $p(\mathbf{z}_0|\mathbf{z})$ is also unknown thus it is again unclear how we can construct a noisy estimate of $\mathbf{z}$. By training a model end-to-end where $\mathbf{z}_0$ is chosen as the origin, however, a prior is implicitly learned over $\mathbf{z}$ such that it is reachable from $\mathbf{z}_0$. Although $p(\mathbf{z})$ is unknown, we do not wish to impose a prior on $\mathbf{z}_0$; the term which enforces this is in Equation 7 is $\log p(\mathbf{z}_0)$, so we can safely ignore this term and simply maximise the likelihood of the data given $\mathbf{z}_0$. Our estimator of $\mathbf{z}$ can therefore be defined simply as $\hat{\mathbf{z}}_{\mathbf{x}}(\mathbf{z}_0) = \mathbf{z}_0 + \nabla_{\mathbf{z}_0} \log p(\mathbf{x}|\mathbf{z}_0)$, which can otherwise be interpreted as a single gradient descent step on the conditional log-likelihood of the data. From this estimate, the data can be reconstructed by passing $\hat{\mathbf{z}}_{\mathbf{x}}$ through the decoder to parameterise $p(\mathbf{x}|\hat{\mathbf{z}}_{\mathbf{x}})$. This procedure can be viewed more explicitly when using a neural network $F \colon \mathbb{R}^k \to \mathbb{R}^m$ to output the mean of $p(\mathbf{x}|\hat{\mathbf{z}}_{\mathbf{x}})$ parameterised by a normal distribution; in this case the

loss function is defined in terms of mean squared error loss $\mathcal{L}^{\text{MSE}}$:

$$G_{\mathbf{x}} = \mathcal{L}^{\text{MSE}}(\mathbf{x}, F(-\nabla_{\mathbf{z}_0}\mathcal{L}^{\text{MSE}}(\mathbf{x}, F(\mathbf{z}_0)))). \tag{8}$$

The gradient computation thereby plays a similar role to an encoder, while $F$ can be viewed as a decoder, with the outer loss term determining the overall reconstruction quality. Using a single network to perform both roles has the advantage of simplifying the overall architecture, avoiding the need to balance networks, and avoiding bottlenecks; this is demonstrated in Figure 1b which provides a visualisation of the GON process.

## 3.3 VARIATIONAL GONS

The variational approach can be naturally applied to GONs, allowing sampling in a single step while only requiring the generative network, reducing the parameters necessary. Similar to before, a feedforward neural network $F$ parameterises $p(\mathbf{x}|\mathbf{z})$, while the expectation of $p(\mathbf{z}|\mathbf{x})$ is calculated with empirical Bayes. A normal prior is assumed over $p(\mathbf{z})$ thus Equation 7 can be written as:

$$\hat{\mathbf{z}}_{\mathbf{x}}(\mathbf{z}_0) = \mathbf{z}_0 + \nabla_{\mathbf{z}_0}\Big(\log p(\mathbf{x}|\mathbf{z}_0) + \log\mathcal{N}(\mathbf{z}_0; \mathbf{0}, 2\boldsymbol{I}_d)\Big), \tag{9}$$

where $\mathbf{z}_0 = \mathbf{0}$ as discussed in Section 3.1. While it would be possible to use this estimate directly within a constant-variance VAE (Ghosh et al., 2019), we opt to incorporate the reparameterisation trick into the generative network as a stochastic layer, to represent the distribution over which $\mathbf{x}$ could be encoded to, using empirical Bayes to estimate $\mathbf{z}$. Similar to the autoencoding approach, we could ignore $\log p(\mathbf{z}_0)$, however we find assuming a normally distributed $\mathbf{z}_0$ implicitly constrains $\mathbf{z}$, aiding the optimisation procedure. Specifically, the forward pass of $F$ is implemented as follows: $\hat{\mathbf{z}}_{\mathbf{x}}$ (or $\mathbf{z}_0$) is mapped by linear transformations to $\mu(\hat{\mathbf{z}}_{\mathbf{x}})$ and $\sigma(\hat{\mathbf{z}}_{\mathbf{x}})$ and the reparameterisation trick is applied, subsequently the further transformations formerly defined as $F$ in the GON formulation are applied, providing parameters for for $p(\mathbf{x}|\hat{\mathbf{z}}_{\mathbf{x}})$. Training is performed end-to-end, minimising the ELBO:

$$\mathcal{L}^{\text{VAE}} = -D_{\text{KL}}(\mathcal{N}(\mu(\hat{\mathbf{z}}_{\mathbf{x}}), \sigma(\hat{\mathbf{z}}_{\mathbf{x}})^2)||\mathcal{N}(\mathbf{0}, \boldsymbol{I}_d)) + \log p(\mathbf{x}|\hat{\mathbf{z}}_{\mathbf{x}}). \tag{10}$$

These steps are shown in Figure 1d, which in practice has a simple implementation.

## 3.4 IMPLICIT GONS

In the field of implicit representation networks, the aim is to learn a neural approximation of a function $\Phi$ that satisfies an implicit equation of the form:

$$R(\mathbf{c}, \Phi, \nabla_\Phi, \nabla_\Phi^2, \dots) = 0, \quad \Phi\colon \mathbf{c} \mapsto \Phi(\mathbf{c}), \tag{11}$$

where $R$'s definition is problem dependent but often corresponds to a loss function. Equations with this structure arise in a myriad of fields, namely 3D modelling, image, video, and audio representation (Sitzmann et al., 2020b). In these cases, data samples $\mathbf{x} = \{(\mathbf{c}, \Phi_{\mathbf{x}}(\mathbf{c}))\}$ can be represented in this form in terms of coordinates $\mathbf{c} \in \mathbb{R}^n$ and the corresponding data at those points $\Phi\colon \mathbb{R}^n \to \mathbb{R}^m$. Due to the continuous nature of $\Phi$, data with large spatial dimensions can be represented much more efficiently than approaches using convolutions, for instance. Despite these benefits, representing a distribution of data points using implicit methods is more challenging, leading to the use of hypernetworks which estimate the weights of an implicit network for each data point (Sitzmann et al., 2020a); this increases the number of parameters and adds significant complexity.

By applying the GON procedure to implicit representation networks, it is possible learn a space of implicit functions without the need for additional networks. We assume there exist latent vectors $\mathbf{z} \in \mathbb{R}^k$ corresponding to data samples; the concatentation of these latent variables with the data coordinates can therefore geometrically be seen as points on a manifold that describe the full dataset in keeping with the manifold hypothesis (Fefferman et al., 2016). An implicit Gradient Origin Network can be trained on this space to mimic $\Phi$ using a neural network $F\colon \mathbb{R}^{n+k} \to \mathbb{R}^m$, thereby learning a space of implicit functions by modifying Equation 8:

$$I_{\mathbf{x}} = \int \mathcal{L}\Big(\Phi_{\mathbf{x}}(\mathbf{c}), F\Big(\mathbf{c} \oplus -\nabla_{\mathbf{z}_0}\int \mathcal{L}\big(\Phi_{\mathbf{x}}(\mathbf{c}), F(\mathbf{c} \oplus \mathbf{z}_0)\big)d\mathbf{c}\Big)\Big)d\mathbf{c}, \tag{12}$$

where both integrations are performed over the space of coordinates and $\mathbf{c} \oplus \mathbf{z}$ represents a concatenation (Figure 1c). Similar to the non-implicit approach, the computation of latent vectors can be expressed as $\mathbf{z} = -\nabla_{\mathbf{z}_0}\int \mathcal{L}\big(\Phi_{\mathbf{x}}(\mathbf{c}), F(\mathbf{c} \oplus \mathbf{z}_0)\big)d\mathbf{c}$. In particular, we parameterise $F$ with a SIREN (Sitzmann et al., 2020b), finding that it is capable of modelling both high and low frequency components in high dimensional spaces.

### 3.5 GON GENERALISATIONS

There are a number of interesting generalisations that make this approach applicable to other tasks. In Equations 8 and 12 we use the same $\mathcal{L}$ in both the inner term and outer term, however, as with variational GONs, it is possible to use different functions; through this, training a GON concurrently as a generative model and classifier is possible, or through some minor modifications to the loss involving the addition of the categorical cross entropy loss function $\mathcal{L}^{\text{CCE}}$ to maximise the likelihood of classification, solely as a classifier:

$$C_{\mathbf{x}} = \mathcal{L}^{\text{CCE}}(f(-\nabla_{\mathbf{z}_0}\mathcal{L}(\mathbf{x}, F(\mathbf{z}_0))), \mathbf{y}), \tag{13}$$

where $\mathbf{y}$ are the target labels and $f$ is a single linear layer followed by softmax. Another possibility is modality conversion for translation tasks; in this case the inner reconstruction loss is performed on the source signal and the outer loss on the target signal.

### 3.6 JUSTIFICATION

Beyond empirical Bayes, we provide some additional analysis on why a single gradient step is in general sufficient as an encoder. Firstly, the gradient of the loss inherently offers substantial information about data making it a good encoder. Secondly, a good latent space should satisfy local consistency (Zhou et al., 2003; Kamnitsas et al., 2018). GONs satisfy this since similar data points will have similar gradients due to the constant latent initialisation. As such, the network needs only to find an equilibrium where its prior is the gradient operation, allowing for significant freedom. Finally, since GONs are trained by backpropagating through empirical Bayes, there are benefits to using an activation function whose second derivative is non-zero.

## 4 RESULTS

We evaluate Gradient Origin Networks on a variety of image datasets: MNIST (LeCun et al., 1998), Fashion-MNIST (Xiao et al., 2017), Small NORB (LeCun et al., 2004), COIL-20 (Nane et al., 1996), CIFAR-10 (Krizhevsky et al., 2009), CelebA (Liu et al., 2015), and LSUN Bedroom (Yu et al., 2015). Simple models are used: for small images, implicit GONs consist of approximately 4 hidden layers of 256 units and convolutional GONs consist of 4 convolution layers with Batch Normalization (Ioffe & Szegedy, 2015) and the ELU non-linearity (Clevert et al., 2016), for larger images the same general architecture is used, scaled up with additional layers; all training is performed with the Adam optimiser (Kingma & Ba, 2015). Error bars in graphs represent standard deviations over three runs.

|  | MNIST | Fashion-MNIST | Small NORB | COIL20 | CIFAR-10 |
|---|---|---|---|---|---|
| Single Step |  |  |  |  |  |
| GON (ours) | **0.41±0.01** | **1.00±0.01** | **0.17±0.01** | 5.35±0.01 | **9.12±0.03** |
| AE | 1.33±0.02 | 1.75±0.02 | 0.73±0.01 | 6.05±0.11 | 12.24±0.05 |
| Tied AE | 2.16±0.03 | 2.45±0.02 | 0.93±0.03 | 6.68±0.13 | 14.12±0.34 |
| 1 Step Detach | 8.17±0.15 | 7.76±0.21 | 1.84±0.01 | 15.72±0.70 | 30.17±0.29 |
| Multiple Steps |  |  |  |  |  |
| 10 Step Detach | 8.13±0.50 | 7.22±0.92 | 1.78±0.02 | 15.48±0.60 | 28.68±1.16 |
| 10 Step | 0.42±0.01 | 1.01±0.01 | **0.17±0.01** | 5.36±0.01 | 9.19±0.01 |
| GLO | 0.70±0.01 | 1.78±0.01 | 0.61±0.01 | **3.27±0.02** | **9.12±0.03** |

Table 1: Validation reconstruction loss (summed squared error) over 500 epochs. For GLO, latents are assigned to data points and jointly optimised with the network. GONs significantly outperform other single step methods and achieve the lowest reconstruction error on four of the five datasets.

|  | MNIST | Fashion-MNIST | Small NORB | COIL20 | CIFAR-10 | CelebA |
|---|---|---|---|---|---|---|
| VGON (ours) | **1.06** | 3.30 | **2.34** | **3.44** | **5.85** | **5.41** |
| VAE | 1.15 | **3.23** | 2.54 | 3.63 | 5.94 | 5.59 |

Table 2: Validation ELBO in bits/dim over 1000 epochs (CelebA is trained over 150 epochs).

### 4.1 QUANTITATIVE EVALUATION

A quantitative evaluation of the representation ability of GONs is performed in Table 1 against a number of baseline approaches. We compare against the single step methods: standard autoencoder, an autoencoder with tied weights (Gedeon, 1998), and a GON with gradients detached from $\mathbf{z}$, as well as the multi-step methods: 10 gradient descent steps per data point, with and without detaching gradients, and a GLO (Bojanowski et al., 2018) which assigns a persistent latent vector to each data point and optimises them with gradient descent, therefore taking orders of magnitude longer to reconstruct validation data than other approaches. For the 10-step methods, a learning rate of 0.1 is applied as used in other literature (Nijkamp et al., 2019); the GLO is trained with MSE for consistency with the other approaches and we do not project latents onto a hypersphere as proposed by the authors since in this experiment sampling is unimportant and this would handicap the approach. GONs achieve much lower validation loss than other single step methods and are competitive with the multi-step approaches; in fact, GONs achieve the lowest loss on four out of the five datasets.

Our variational GON is compared with a VAE, whose decoder is the same as the GON, quantitatively in terms of ELBO on the test set in Table 2. We find that the representation ability of GONs is pertinent here also, allowing the variational GON to achieve lower ELBO on five out of the six datasets tested. Both models were trained with the original VAE formulation for fairness, however, we note that variations aimed at improving VAE sample quality such as $\beta$-VAEs (Higgins et al., 2017) are also applicable to variational GONs to the same effect.

An ablation study is performed, comparing convolutional GONs with autoencoders whose decoders have exactly the same architecture as the GONs (Figure 2a) and where the autoencoder decoders mirror their respective encoders. The mean reconstruction loss over the test set is measured after each epoch for a variety of latent sizes. Despite having half the number of parameters and linear encodings,

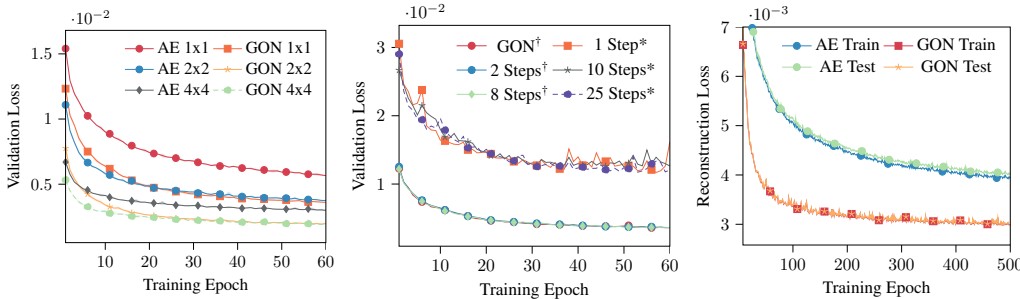

(a) GONs achieve lower validation loss than autoencoders.

(b) Multiple latent update steps. *=grads detached, †=not detached.

(c) GONs overfit less than standard autoencoders.

Figure 2: Gradient Origin Networks trained on CIFAR-10 are found to outperform autoencoders using exactly the same architecture without the encoder, requiring half the number of parameters.

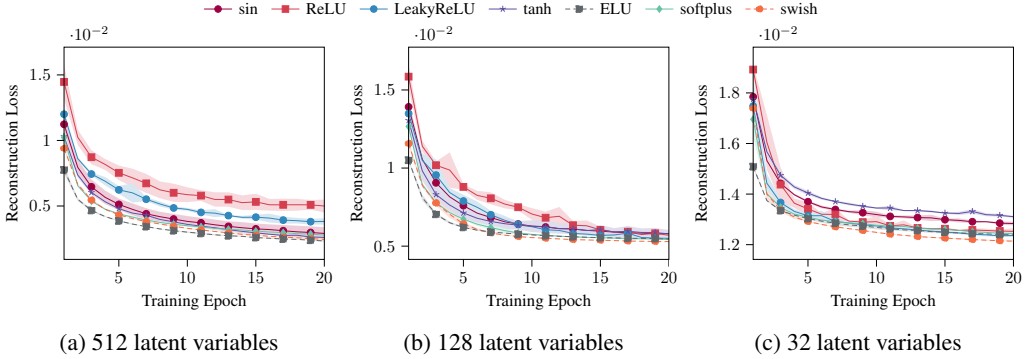

(a) 512 latent variables

(b) 128 latent variables

(c) 32 latent variables

Figure 3: The impact of activation function and number of latent variables on model performance for a GON measured by comparing reconstruction losses through training.

GONs achieve significantly lower reconstruction loss over a wide range of architectures. Further experiments evaluating the impact of latent size and model capacity are performed in Appendix B.

Our hypothesis that for high dimensional datasets, a single gradient step is sufficient when jointly optimised with the forwards pass is tested on the CIFAR-10 dataset in Figure 2b. We observe negligible difference between a single step and multiple jointly optimised steps, in support of our hypothesis. Performing multiple steps greatly increases run-time so there is seemingly no benefit in this case. Additionally, the importance of the joint optimisation procedure is determined by detaching the gradients from **z** before reconstructing (Figure 2b); this results in markedly worse performance, even when in conjunction with multiple steps. In Figure 2c we assess whether the greater performance of GONs relative to autoencoders comes at the expense of generalisation; we find that the opposite is true, that the discrepancy between reconstruction loss on the training and test sets is greater with autoencoders. This result extends to other datasets as can be seen in Appendix E.

Figure 3 demonstrates the effect activation functions have on convolutional GON performance for different numbers of latent variables. Since optimising GONs requires computing second order derivatives, the choice of nonlinearity requires different characteristics to standard models. In particular, GONs prefer functions that are not only effective activation functions, but also whose second derivatives are non-zero, unlike ReLUs where $\text{ReLU}''(x) = 0$. The ELU non-linearity is effective with all tested architectures.

## 4.2 QUALITATIVE EVALUATION

The representation ability of implicit GONs is shown in Figure 4 where we train on large image datasets using a relatively small number of parameters. In particular, Figure 4a shows MNIST can be well-represented with just 4,385 parameters (a SIREN with 3 hidden layers each with 32 hidden units, and 32 latent dimensions). An advantage of modelling data with implicit networks is that coordinates can be at arbitrarily high resolutions. In Figure 5 we train on 32x32 images then reconstruct at 256x256. A significant amount of high frequency detail is modelled despite only seeing low resolution images. The structure of the implicit GON latent space is shown by sampling latent codes from pairs of images, and then spherically interpolating between them to synthesise new samples (Figure 6). These samples are shown to capture variations in shape (the shoes in Figure 6b), size, and rotation (Figure 6d).

We assess the quality of samples from variational GONs using convolutional models trained to convergence in Figure 7. These are diverse and often contain fine details. Samples from an implicit GON trained with early stopping, as a simple alternative, can be found in Appendix D however this approach results in fine details being lost. GONs are also found to converge quickly; we plot

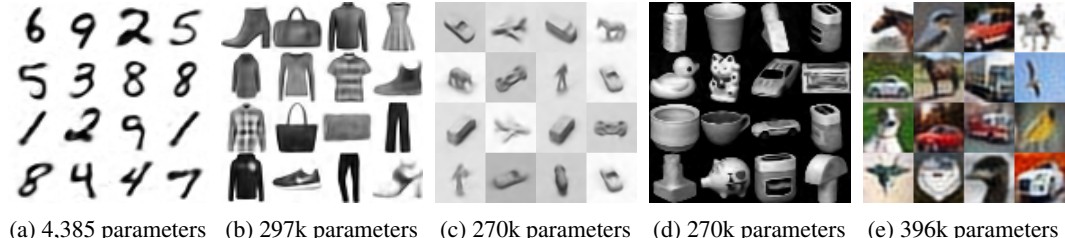

(a) 4,385 parameters  (b) 297k parameters  (c) 270k parameters  (d) 270k parameters  (e) 396k parameters

Figure 4: Training implicit GONs with few parameters demonstrates their representation ability.

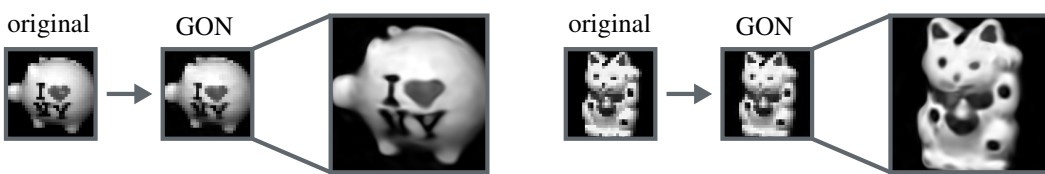

Figure 5: By training an implicit GON on 32x32 images, then sampling at 256x256, super-resolution is possible despite never observing high resolution data.

reconstructions at multiple time points during the first minute of training (Figure 8). After only 3 seconds of training on a single GPU, a large amount of signal information from MNIST is modelled.

In order to evaluate how well GONs can represent high resolution natural images, we train a convolutional GON on the LSUN Bedroom dataset scaled to 128x128 (Figure 9a). As with smaller, more simple data, we find training to always be extremely stable and consistent over a wide range of hyperparameter settings. Reconstructions are of excellent quality given the simple network architecture. A convolutional variational GON is also trained on the CelebA dataset scaled to 64x64 (Figure 9b). Unconditional samples are somewhat blurry as commonly associated with traditional VAE models on complex natural images (Zhao et al., 2017) but otherwise show wide variety.

## 5 DISCUSSION

Despite similarities with autoencoder approaches, the absence of an encoding network offers several advantages. VAEs with overpowered decoders are known to ignore the latent variables (Chen et al., 2017) whereas GONs only have one network that equally serves both encoding and decoding functionality. Designing inference networks for complicated decoders is not a trivial task (Vahdat & Kautz, 2020), however, inferring latent variables using a GON simplifies this procedure. Similar to GONs, Normalizing Flow methods are also capable of encoding and decoding with a single

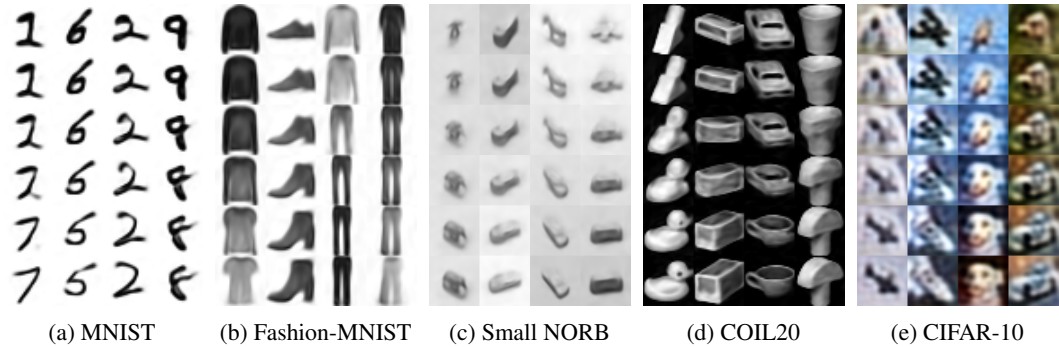

| (a) MNIST | (b) Fashion-MNIST | (c) Small NORB | (d) COIL20 | (e) CIFAR-10 |

Figure 6: Spherical linear interpolations between points in the latent space for trained implicit GONs using different datasets (approximately 2-10 minutes training per dataset on a single GPU).

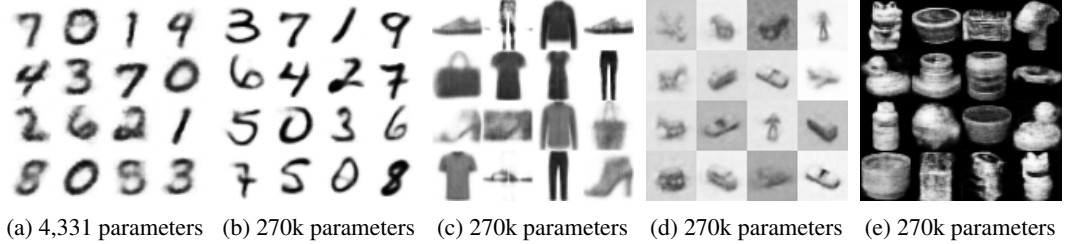

| (a) 4,331 parameters | (b) 270k parameters | (c) 270k parameters | (d) 270k parameters | (e) 270k parameters |

Figure 7: Random samples from a convolutional variational GON with normally distributed latents.

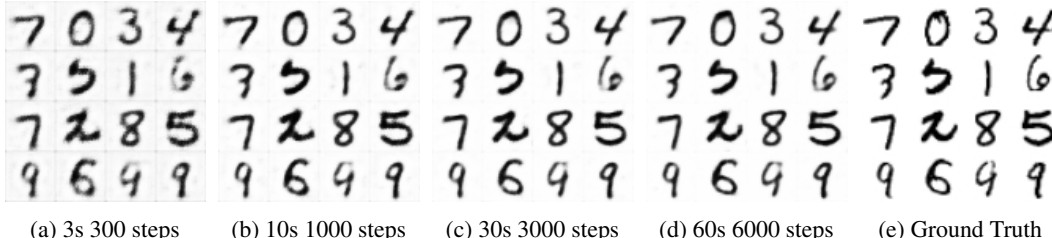

| (a) 3s 300 steps | (b) 10s 1000 steps | (c) 30s 3000 steps | (d) 60s 6000 steps | (e) Ground Truth |

Figure 8: Convergence of convolutional GONs with 74k parameters.

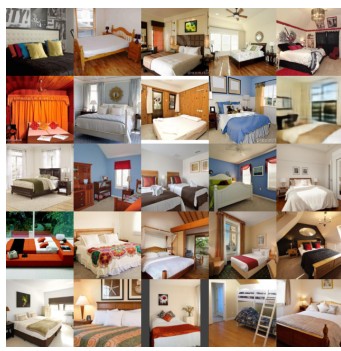 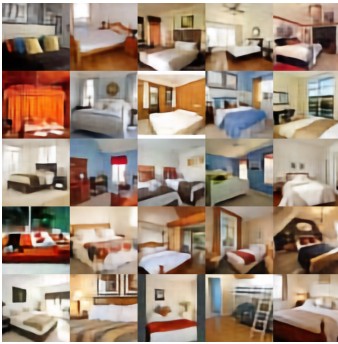 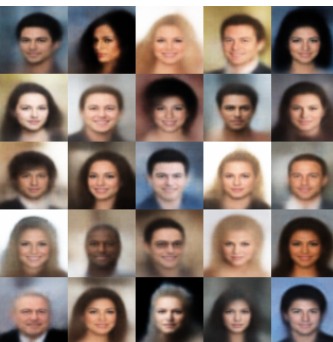

(a) LSUN 128x128 Bedroom validation images (left) reconstructed by a convolutional GON (right).

(b) Samples from a convolutional variational GON trained on CelebA.

Figure 9: GONs are able to represent high resolution complex datasets to a high degree of fidelity.

set of weights, however, they achieve this by restricting the network to be invertible. This requires considerable architectural restrictions that affect performance, make them less parameter efficient, and are unable to reduce the dimension of the data (Kingma & Dhariwal, 2018). Similarly, autoencoders with tied weights also encode and decode with a single set of weights by using the transpose of the encoder's weight matrices in the decoder; this, however, is only applicable to simple architectures. GONs on the other hand use gradients as encodings which allow arbitrary functions to be used.

A number of previous works have used gradient-based computations to learn latent vectors however as far as we are aware, we are the first to use a single gradient step jointly optimised with the feedforward pass, making it fundamentally different to these approaches. Latent encodings have been estimated for pre-trained generative models without encoders, namely Generative Adversarial Networks, using approaches such as standard gradient descent (Lipton & Tripathi, 2017; Zhu et al., 2016). A number of approaches have trained generative models directly with gradient-based inference (Han et al., 2017; Bojanowski et al., 2018; Zadeh et al., 2019); these assign latent vectors to data points and jointly learn them with the network parameters through standard gradient descent or Langevin dynamics. This is very slow, however, and convergence for unseen data samples is not guaranteed. Short run MCMC has also been applied (Nijkamp et al., 2020) however this still requires approximately 25 update steps. Since GONs train end-to-end, the optimiser can make use of the second order derivatives to allow for inference in a single step. Also of relevance is model-agnostic meta-learning (Finn et al., 2017), which trains an architecture so that a few gradient descent steps are all that are necessary to new tasks. This is achieved by backpropagating through these gradients, similar to GONs.

In the case of implicit GONs, the integration terms in Equation 12 result in computation time that scales in proportion with the data dimension. This makes training slower for very high dimensional data, although we have not yet investigated Monte Carlo integration. In general GONs are stable and consistent, capable of generating quality samples with an exceptionally small number of parameters, and converge to diverse results with few iterations. Nevertheless, there are avenues to explore so as to improve the quality of samples and scale to larger datasets. In particular, it would be beneficial to focus on how to better sample these models, perform formal analysis on the gradients, and investigate whether the distance function could be improved to better capture fine details.

## CONCLUSION

In conclusion, we have introduced a method based on empirical Bayes which computes the gradient of the data fitting loss with respect to the origin, and then jointly fits the data while learning this new point of reference in the latent space. The results show that this approach is able to represent datasets using a small number of parameters with a simple overall architecture, which has advantages in applications such as implicit representation networks. GONs are shown to converge faster with lower overall reconstruction loss than autoencoders, using the exact same architecture but without the encoder. Experiments show that the choice of non-linearity is important, as the network derivative jointly acts as the encoding function.

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

## A PROOF OF CONDITIONAL EMPIRICAL BAYES

For latent variables $\mathbf{z} \sim p_{\mathbf{z}}$, a noisy observation of $\mathbf{z}$, $\mathbf{z}_0 = \mathbf{z} + \mathcal{N}(\mathbf{0}, \boldsymbol{I}_d)$, and data point $\mathbf{x} \sim p_d$, we wish to find an estimator of $p(\mathbf{z}|\mathbf{x})$. To achieve this, we condition the Bayes least squares estimator on $\mathbf{x}$:

$$\hat{\mathbf{z}}_{\mathbf{x}}(\mathbf{z}_0) = \int \mathbf{z} p(\mathbf{z}|\mathbf{z}_0, \mathbf{x}) d\mathbf{z} = \int \mathbf{z} \frac{p(\mathbf{z}_0, \mathbf{z}|\mathbf{x})}{p(\mathbf{z}_0|\mathbf{x})} d\mathbf{z}. \tag{14}$$

Through the definition of the probabilistic chain rule and by marginalising out $\mathbf{z}$, we can define $p(\mathbf{z}_0|\mathbf{x}) = \int p(\mathbf{z}_0|\mathbf{z}, \mathbf{x}) p(\mathbf{z}|\mathbf{x}) d\mathbf{z}$ which can be simplified to $\int p(\mathbf{z}_0|\mathbf{z}) p(\mathbf{z}|\mathbf{x}) d\mathbf{z}$ since $\mathbf{z}_0$ is dependent only on $\mathbf{z}$. Writing this out fully, we obtain:

$$p(\mathbf{z}_0|\mathbf{x}) = \int \frac{1}{(2\pi)^{d/2} |\det(\boldsymbol{\Sigma})|^{1/2}} \exp\left(-(\mathbf{z}_0 - \mathbf{z})^T \boldsymbol{\Sigma}^{-1}(\mathbf{z}_0 - \mathbf{z})/2\right) p(\mathbf{z}|\mathbf{x}) d\mathbf{z}. \tag{15}$$

Differentiating with respect to $\mathbf{z}_0$ and multiplying both sides by $\mathbf{\Sigma}$ gives:

$$\mathbf{\Sigma}\nabla_{\mathbf{z}_0}p(\mathbf{z}_0|\mathbf{x}) = \int(\mathbf{z}-\mathbf{z}_0)p(\mathbf{z}_0|\mathbf{z},\mathbf{x})p(\mathbf{z}|\mathbf{x})d\mathbf{z} = \int(\mathbf{z}-\mathbf{z}_0)p(\mathbf{z}_0,\mathbf{z}|\mathbf{x})d\mathbf{z} \qquad (16)$$

$$= \int \mathbf{z}p(\mathbf{z}_0,\mathbf{z}|\mathbf{x})d\mathbf{z} - \mathbf{z}_0 p(\mathbf{z}_0|\mathbf{x}). \qquad (17)$$

After dividing both sides by $p(\mathbf{z}_0|\mathbf{x})$ and combining with Equation 14 we get:

$$\mathbf{\Sigma}\frac{\nabla_{\mathbf{z}_0}p(\mathbf{z}_0|\mathbf{x})}{p(\mathbf{z}_0|\mathbf{x})} = \int \mathbf{z}\frac{p(\mathbf{z}_0,\mathbf{z}|\mathbf{x})}{p(\mathbf{z}_0|\mathbf{x})}d\mathbf{z} - \mathbf{z}_0 = \hat{\mathbf{z}}_{\mathbf{x}}(\mathbf{z}_0) - \mathbf{z}_0. \qquad (18)$$

Finally, this can be rearranged to give the single step estimator of $\mathbf{z}$:

$$\hat{\mathbf{z}}_{\mathbf{x}}(\mathbf{z}_0) = \mathbf{z}_0 + \mathbf{\Sigma}\frac{\nabla_{\mathbf{z}_0}p(\mathbf{z}_0|\mathbf{x})}{p(\mathbf{z}_0|\mathbf{x})} = \mathbf{z}_0 + \mathbf{\Sigma}\nabla_{\mathbf{z}_0}\log p(\mathbf{z}_0|\mathbf{x}). \qquad (19)$$

## B    COMPARISON WITH AUTOENCODERS

This section compares Gradient Origin Networks with autoencoders that use exactly the same architecture, as described in the paper. All experiments are performed with the CIFAR-10 dataset. After each epoch the model is fixed and the mean reconstruction loss is measured over the training set.

In Figure 10a the depth of networks is altered by removing blocks thereby downscaling to various resolutions. We find that GONs outperform autoencoders until latents have a spatial size of 8x8 (where the equivalent GON now only has only 2 convolution layers). Considering the limit where neither model has any parameters, the latent space is the input data i.e. $\mathbf{z} = \mathbf{x}$. Substituting the definition of a GON (Equation 8) this becomes $-\nabla_{\mathbf{z}_0}\mathcal{L} = \mathbf{x}$ which simplifies to $\mathbf{0} = \mathbf{x}$ which is a contradiction. This is not a concern in normal practice, as evidenced by the results presented here.

Figure 10b explores the relationship between autoencoders and GONs when changing the number of convolution filters; GONs are found to outperform autoencoders in all cases. The discrepancy between loss curves decreases as the number of filters increases likely due to the diminishing returns of providing more capacity to the GON when its loss is significantly closer to 0.

A similar pattern is found when decreasing the latent space (Figure 10c); in this case the latent space likely becomes the limiting factor. With larger latent spaces GONs significantly outperform autoencoders, however, when the latent bottleneck becomes more pronounced this lead lessens.

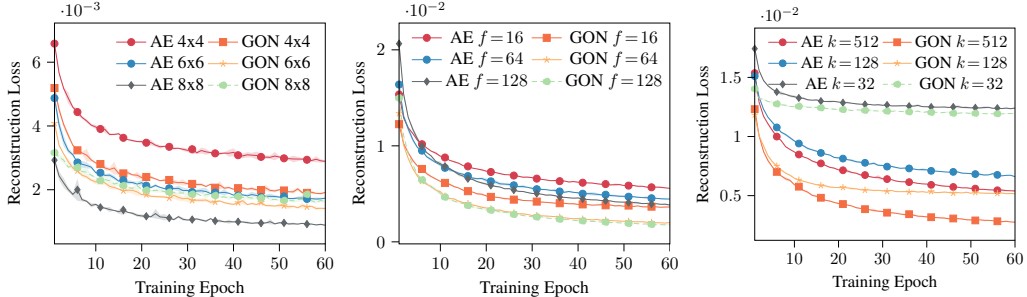

(a) Encoding images to various spatial dimensions.

(b) Varying capacities by changing the number of convolution filters $f$.

(c) Encoding images to a variety of latent space sizes.

Figure 10: Experiments comparing convolutional GONs with autoencoders on CIFAR-10, where the GON uses exactly same architecture as the AE, without the encoder. (a) At the limit autoencoders tend towards the identity function whereas GONs are unable to operate with no parameters. As the number of network parameters increases (b) and the latent size decreases (c), the performance lead of GONs over AEs decreases due to diminishing returns/bottlenecking.

## C  INITIALISING $\mathbf{z}_0$ AT THE ORIGIN

We evaluate different initialisations of $\mathbf{z}_0$ in Figure 11 by sampling $\mathbf{z}_0 \sim \mathcal{N}(\mathbf{0}, \sigma^2 \boldsymbol{I})$ for a variety of standard deviations $\sigma$. The proposed approach ($\sigma = 0$) achieves the lowest reconstruction loss (Figure 11a); results for $\sigma > 0$ are similar, suggesting that the latent space is adjusted so $\mathbf{z}_0$ simulates the origin. An alternative parameterisation of $\mathbf{z}$ is to use a use a single gradient descent style step $\mathbf{z} = \mathbf{z}_0 - \nabla_{\mathbf{z}_0}\mathcal{L}$ (Figure 11b), however, losses are higher than the proposed GON initialisation.

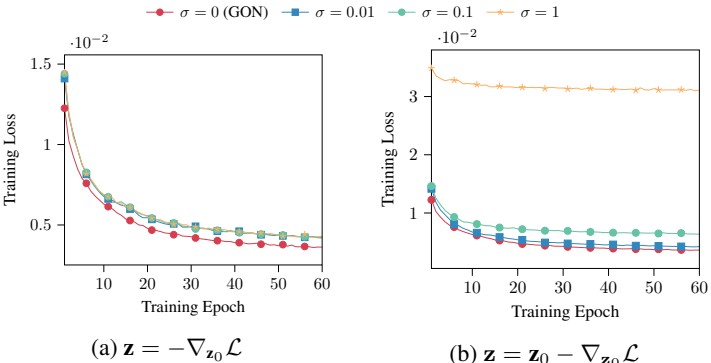

(a) $\mathbf{z} = -\nabla_{\mathbf{z}_0}\mathcal{L}$  (b) $\mathbf{z} = \mathbf{z}_0 - \nabla_{\mathbf{z}_0}\mathcal{L}$

Figure 11: Training GONs with $\mathbf{z}_0$ sampled from a variety of normal distributions with different standard deviations $\sigma$, $\mathbf{z}_0 \sim \mathcal{N}(\mathbf{0}, \sigma^2 \boldsymbol{I})$. Approach (a) directly uses the negative gradients as encodings while approach (b) performs one gradient descent style step initialised at $\mathbf{z}_0$.

## D  LATENT SPACE AND EARLY STOPPING

Figure 12 shows a GON trained as a classifier where the latent space is squeezed into 2D for visualisation (Equation 13) and Figure 13 shows that the negative gradients of a GON after 800 steps of training are approximately normally distributed; Figure 14 shows that the latents of a typical convolutional VAE after 800 steps are significantly less normally distributed than GONs.

To obtain new samples with implicit GONs, we can use early stopping as a simple alternative to variational approaches. These samples (Figure 15) are diverse and capture the object shapes.

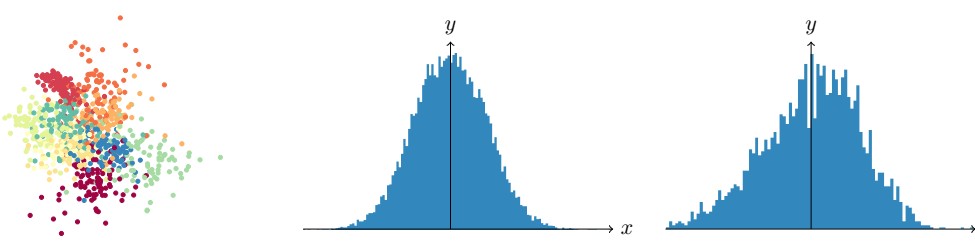

Figure 12: 2D latent space samples of an implicit GON classifier trained on MNIST (class colours).

Figure 13: Histogram of latent gradients after 800 implicit GON steps with a SIREN.

Figure 14: Histogram of traditional VAE latents after 800 steps.

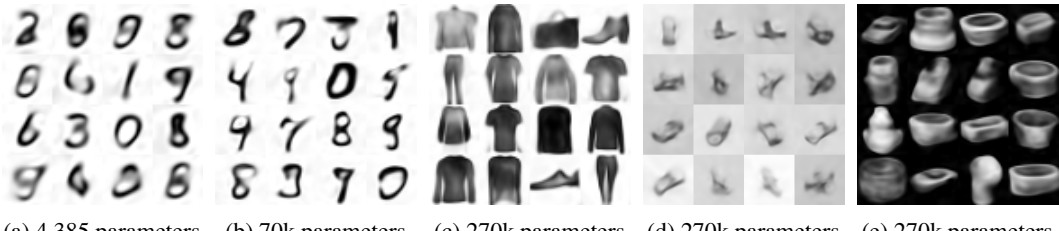

(a) 4,385 parameters    (b) 70k parameters    (c) 270k parameters    (d) 270k parameters    (e) 270k parameters

Figure 15: GONs trained with early stopping can be sampled by approximating their latent space with a multivariate normal distribution. These images show samples from an implicit GON trained with early stopping.

# E  GON GENERALISATION

In Figure 16 the training and test losses for GONs and autoencoders are plotted for a variety of datasets. In all cases, GONs generalise better than their equivalent autoencoders while achieving lower losses with fewer parameters.

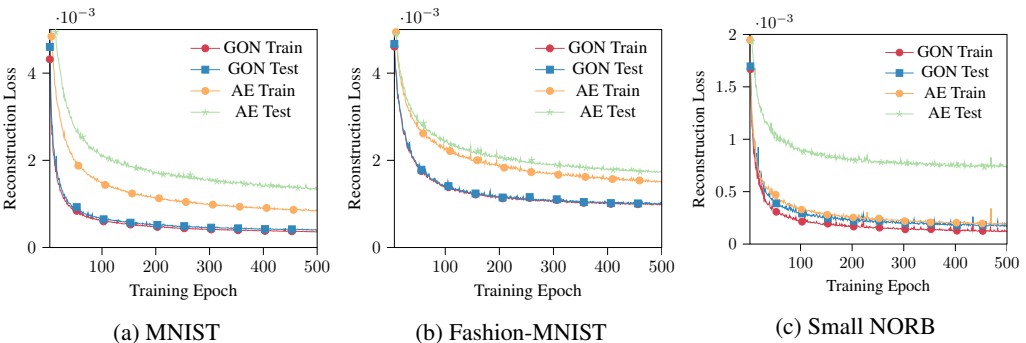

(a) MNIST          (b) Fashion-MNIST          (c) Small NORB

Figure 16: The discrepancy between training and test reconstruction losses when using a GON is smaller than equivalent autoencoders over a variety of datasets.

# F  AVAILABILITY

Source code for the convolutional GON, variational GON, and implicit GON is available under the MIT license on GitHub at: `https://github.com/cwkx/GON`. This implementation uses PyTorch and all reported experiments use a Nvidia RTX 2080 Ti GPU.

## G    SUPER-RESOLUTION

Additional super-sampled images obtained by training an implicit GON on 28x28 MNIST data then reconstructing test data at a resolution of 256x256 are shown in Figure 17.

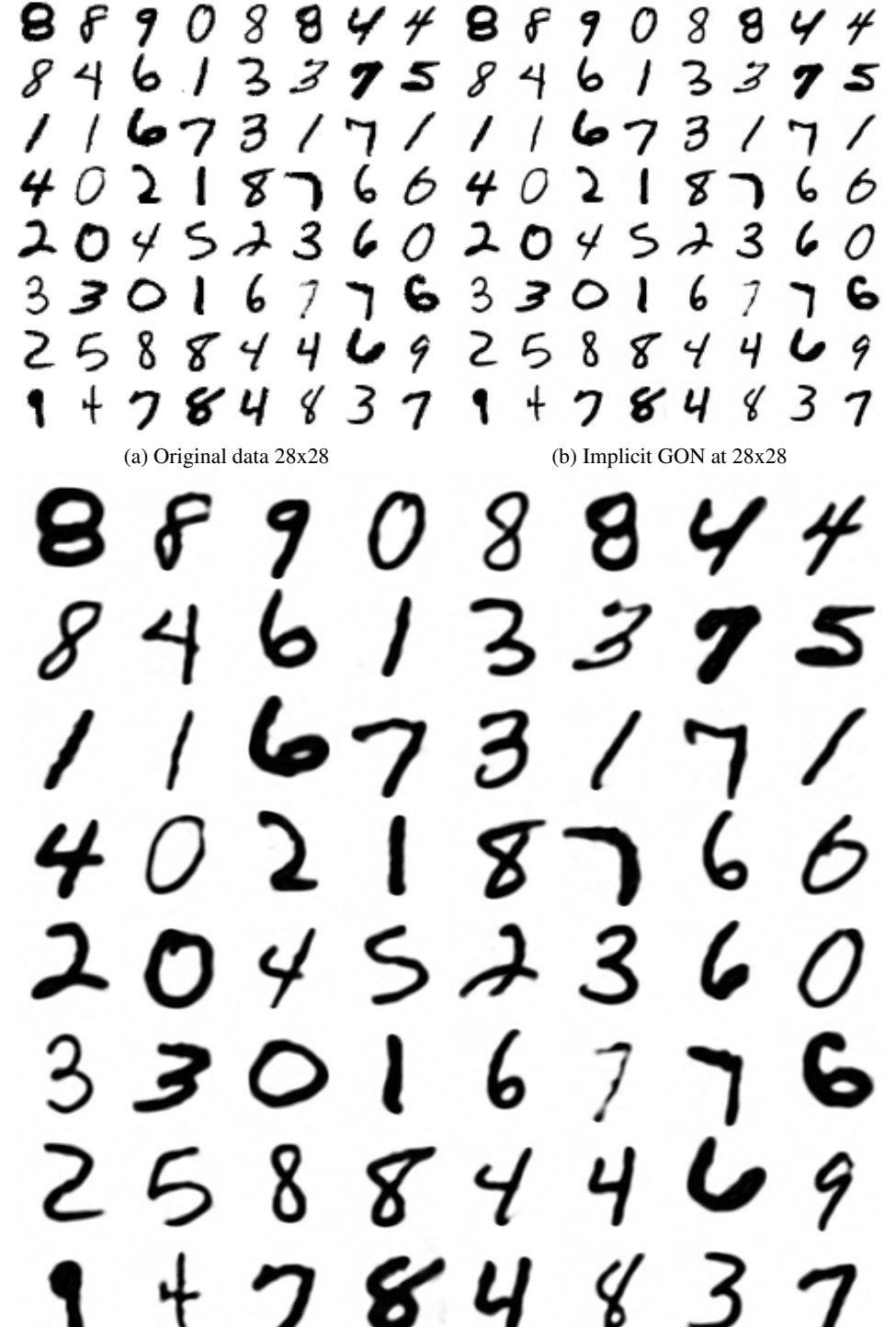

(a) Original data 28x28                    (b) Implicit GON at 28x28

Figure 17: Super-sampling 28x28 MNIST test data at 256x256 coordinates using an implicit GON.

