# OpenReview forum: "Gradient Origin Networks"
_ICLR.cc/2021/Conference — ICLR 2021 Poster_

### Official Review · AnonReviewer1 · 2020-10-20
**Very interesting paper and findings, but seems somewhat rushed.**

**Rating:** 7
**Confidence:** 4

**Review:**

This paper introduces a "new" inference method for autoencoder-type models, where the encoder is taken as a gradient of the decoder with respect to a zero-initialized latent variable. The method is evaluated for both a deterministic autoencoder and a VAE on toy image data (cifar10 being the most complex of them) and applied to convolutional decoder and to SIREN-type implicit representation networks. This is, for all intents and purposes, a single step iterative inference setup. In its VAE variant it is extremely similar to old-school iterative inference, albeit with a single gradient step.

The paper is very-well written and interesting. The method seems to be getting very good results,. Still, the paper seems to be rushed. The results are only on small scale and toyish datasets, and there are very few baselines.

In its current state I recommend rejection due to rather limited novelty (although it's cool to see that this type of inference works for implicit scene representations) and very limited evaluation. There are also very many links to existing literature that are not properly described. Let me elaborate.

Baselines:
To determine the efficacy of this method, the authors would have to compare against some similar methods including:
* old-school multi-step variational inference
* semi-amortized variational inference
* the proposed method with multiple gradient steps
* the proposed method with detached gradient (as in: not use 2nd order gradients)
* a fully-convolutional autoencoder with parameters tied between the encoder and decoder. This is for two reasons: a) this would reduce the number of parameters by half, making it more similar to GON, but also b) the transposed-convolution used in such a setup corresponds almost exactly to the gradient of the encoder, which is an idea very similar to GONs.

Missing links to the literature:
* the above fully-conv AE setup.
* model-agnostic meta-learning (and related, e.g. CAVIA, LEO etc), where the "latents" are produced by single- or multi-step optimization.

Missing experiments:
We would need more evidence to determine if such a simple method is useful. A good experiment would be e.g. on imagenet.

Further suggestions:
Subfigures in fig2 and 3 (and most of figs in the appendix) use different scales on the Y axis. It would be easier to read the figures if the scaled were normalized within a single figure.

Update: I've updated the score given the authors' response, see my comment below.

---

> ### Author Response · Authors · 2020-11-20
> **Response to Reviewer 1**
>
> Thank you for your constructive feedback and excellent suggestions. We will address your comments point-by-point:
>
> > “The results are only on small scale and toyish datasets”
>
> In our latest update we have added experiments on larger scale more complex datasets: 128x128 LSUN Bedrooms for the GON, and CelebA 64x64 for the variational GON.
>
> > “Baselines: To determine the efficacy of this method, the authors would have to compare against some similar methods including...”
>
> For our non-variational approach, we have added quantitative comparisons with autoencoders+tied weights, our approach with detached gradients, our approach with multiple gradient descent steps, both with and without detached gradients, as suggested, as well as with a GLO (Bojanowski 2018) which assigns a latent vector to each data point and jointly optimises these with the network parameters (Table 1). We find that GONs achieve the lowest validation losses on 3 of the 5 datasets tested. Notably, all other single step approaches result in significantly high reconstruction loss.
>
> We have also added quantitative comparisons with vanilla VAEs in Table 2, finding GONs to achieve lower ELBO on 5 of the 6 datasets. We aim to add comparisons with other variational approaches as suggested, in another update.
>
> > “b) the transposed-convolution used in such a setup corresponds almost exactly to the gradient of the encoder, which is an idea very similar to GONs.”
>
> While the gradient through a single convolution layer is indeed related to transposed convolutions, when convolutions are composed and/or interleaved with other functions, the gradient becomes much more complex. Indeed, the gradient of deep MLPs corresponds to a product of networks. Additionally, restricting the architecture to using tied-weights is not necessarily applicable to more complex architectures whereas using the gradient can be applied to any function.
>
> > “Missing links to the literature…”
>
> We have now integrated discussion of autoencoders with tied weights and the connections with model-agnostic meta-learning into the Discussion section.
>
> > “Missing experiments: We would need more evidence to determine if such a simple method is useful. A good experiment would be e.g. on imagenet.”
>
> We hope that the aforementioned additional quantitative experiments (Tables 1 and 2) including experiments on CelebA, and qualitative examples on LSUN Bedrooms assuage your concerns. Experiments have also been added to evaluate our claim that a single gradient step is sufficient and that GONs do not memorise datasets in Figures 2b and c respectively.
>
> > “Further suggestions: Subfigures in fig2 and 3 (and most of figs in the appendix) use different scales on the Y axis. It would be easier to read the figures if the scaled were normalized within a single figure.”
>
> All figures which we deemed legible when normalised with a single figure (previously Figures 2, 9, 10, and 11) have been changed as per your suggestion (now Figures 2 and 10).

---

> > ### Comment · AnonReviewer1 · 2020-11-23
> > **Thanks for the edits**
> >
> > Thanks for addressing the majority of my comments, this is a much stronger paper now and I'm happy to increase my score. Some further suggestions:
> > - Table 1 should have the best results in boldface.
> > - It'd be great to have errors bars in Table 1 and 2 as well as most figures.
> > - In sec 3.6 "non-linearity function" should be "non-linear function".
> > - I like the newly-introduced argument (connections to MoE) of why GONs work well, though I must say that I am still baffled by how good it is.

---

> > > ### Author Response · Authors · 2020-11-24
> > > **Response to Reviewer 1**
> > >
> > > Thank you for your kind words and for updating the score. We have implemented these suggested changes; error bars (as envelopes) representing standard deviation have been added to Table 1 and Figures 2a, 3, and 10 however we deemed it inappropriate for the other figures due to significant line overlapping impacting the presentation of these results.

---

### Official Review · AnonReviewer2 · 2020-10-28
**An interesting new perspective on generative modeling and implicit representation learning, but incomplete in its execution.**

**Rating:** 7
**Confidence:** 4

**Review:**

The paper proposes GONs which seek to build a generative model with an “implicit” encoder that comes essentially for free with the use of a few re-parameterization tricks. The main idea being that existing generative models with an encoder are “redundant” in that the decoder itself has the ability to compute the gradient with respect to a latent vector, z, which itself can be thought of as the “encoding”. Since the choice of what initial latent vector to choose arises here, the paper advocates for simply choosing a z_0 which is a zero vector. In addition to the “explicit” formulation, there is also an implicit GON which is proposed that can generalize implicit generative models  (like SIREN) to entire distributions as opposed to a single data point, as they are currently used.

Overall, I think this is very interesting work but incomplete. Considering GONs are a completely new category of generative models, it would greatly help to study each piece in more detail (theoretically or empirically) to establish what makes GONs successful, different, and how this improves our understanding of implicit representations in neural networks.


Strengths:

+ An interesting and novel formulation of encoding schemes from decoders that do not need any additional training or networks.
+ The paper explores several different variants of GONs — from a variational alternative, implicit, and a classifier. Which greatly expands its scope of application in new problems.
+ GONs generalize implicit generative models like SIRENs to work with an entire data distribution with very few parameters, which I think is a great benefit. This also naturally allows for variational alternatives, meaning we can sample from complex high dimensional distributions using very simple networks.
+ The implicit GON also enables finer grid sampling in the input space, enabling its use in applications like super resolution naturally — but to any image from the training distribution.


Weaknesses:

* The paper is very dense in terms of ideas, and as such falls short in thoroughly evaluating all of them. For example, the paper contributes several ideas like GONs, implicit GONs, variational GONs, which is great but it would help if each one of those pieces were studied in some more detail so they can be compared and contextualized better with existing approaches. For example, in the formulation itself the GON loss is presented “as is”, but I think it warrants some more study.
    * For example, why is just a single step “sufficient” to estimate “z”? Does the quality of “z” improve if you take multiple smaller steps? How stable is this for different datasets? The empirical studies show promise, that indeed this can work reasonably well in reconstructing different datasets, but it would greatly help to justify some of these choices further.
    * In the explicit case, how important is the choice of “F” ? The choice of activation function is explored but what about the architecture/ number of parameters for a given dataset?
* In all the experiments, the reconstruction losses are shown are for the training set, how do the validation set samples get reconstructed?  It’s not clear if GONs are so effective in reconstructing because they are memorizing the data?
* How does the performance of GONs change as the size of the output space grows larger? For e.g. 128x128 or 256x256?
* Some of the terminology is also confusing. What does it mean when you “overfit” to an entire distribution? I understand its usage for a single image, but it's not clear what this means for an entire dataset. Are the samples from Figure 4 all from the *same* trained GON?
* Is Figure 7 from an explicit GON or an implicit GON? If its explicit, how are the number of parameters comparable to an implicitGON? Clearly an explicit model will have a lot more number of parameters. esp as the size of the images increase?
* I really like and appreciate the variationalGON experiments. How do they compare with  standard VAEs? Can they recover CelebA 64x64 images? How would they compare on quantitative metrics like FID etc.?
* In the super resolution experiment, can it super resolve *any* image from the distribution it was trained on? For e.g. in figure 5. is it just a matter of resampling the grid to 256x256 and running them through the pre-trained model for any sample from p(x)?

---------- Update on the revised manuscript ----------

I have read the new version of the paper and it reads a lot better. The new expanded methods section, and the definitions for different variations of GONs makes the paper much stronger and easier to understand. I appreciate and like the new experiments that show GONs capabilities on LSUN, comparisons with VAE on ELBO.

Most of my concerns have been addressed in this version. I think this paper makes an interesting and novel contribution and I will raise my score accordingly.

---

> ### Author Response · Authors · 2020-11-20
> **Response to Reviewer 2**
>
> Thank you for your excellent comprehensive feedback. We have incorporated many of your suggestions in the most recent update. We will address your comments point-by-point:
>
> > “The paper is very dense in terms of ideas, and as such falls short in thoroughly evaluating all of them. For example, the paper contributes several ideas like GONs, implicit GONs, variational GONs, which is great but it would help if each one of those pieces were studied in some more detail so they can be compared and contextualized better with existing approaches. For example, in the formulation itself the GON loss is presented “as is”, but I think it warrants some more study. ”
>
> We have greatly expanded the method section, deriving our approach from empirical Bayes. Specifically, there is now a preliminaries section which introduces the concept of empirical Bayes and variational autoencoders in detail; additionally, the method section is now divided into sections, covering our contributions (GON, variational GON, implicit GON, and generalisations) in more detail as well as more thoroughly introducing the surrounding concepts.
>
> > “For example, why is just a single step “sufficient” to estimate “z”? Does the quality of “z” improve if you take multiple smaller steps? How stable is this for different datasets? The empirical studies show promise, that indeed this can work reasonably well in reconstructing different datasets, but it would greatly help to justify some of these choices further. ”
>
> Our new derivation from empirical Bayes shows that if we consider z_0 a noisy approximation of z, then we can use a single gradient step to calculate the expected value of p(z|x). As mentioned in our response to Reviewer 3, we also provide an explanation from a function approximating perspective, namely, the derivative of a deep MLP corresponds to a product of networks, allowing efficient modelling of high dimensional data.
>
> We have added experiments to evaluate the claim that a single step is sufficient in Figure 2b, finding that when jointly optimised, multiple steps offer no notable improvement over a single step, and training with gradients of z detached results in significantly worse performance. This can also be seen over a broad range of datasets, trained over long periods, in Table 1.
>
> In terms of stability, we have observed no issues when training these models. Standard architectures are used and the Adam optimiser with default values. This is the case even on high resolutions data (up to 128x128 images tested). Additionally, we find them to be extremely consistent over multiple runs.
>
> > “In the explicit case, how important is the choice of “F” ? The choice of activation function is explored but what about the architecture/ number of parameters for a given dataset?”
>
> In Figure 10b the effect of number of parameters is explored. We find that GONs outperform their equivalent autoencoder in all cases. As the number of parameters is increased, this lead lessens due to diminishing returns. While we only use simple architectures, we have found all variations (e.g. upsampling, transposed convolutions, instance normalisation) to be effective.
>
> > “In all the experiments, the reconstruction losses are shown are for the training set, how do the validation set samples get reconstructed? It’s not clear if GONs are so effective in reconstructing because they are memorizing the data?”
>
> We have added extra experiments to assess this. In Figure 2c, Table 1, and Figure 16 training and validation losses are plotted for both GONs and their equivalent autoencoder; this demonstrates that GONs not only do not memorise the data, but appear to generalise better than autoencoders.
>
> > “How does the performance of GONs change as the size of the output space grows larger? For e.g. 128x128 or 256x256?”
>
> We find the performance to be on par with smaller sized outputs. This is evaluated qualitatively by reconstructing 128x128 LSUN Bedrooms data with a convolutional GON in Figure 9a where a substantial amount of detail is modelled.
>
> > “Some of the terminology is also confusing. What does it mean when you “overfit” to an entire distribution? I understand its usage for a single image, but it's not clear what this means for an entire dataset. Are the samples from Figure 4 all from the same trained GON?”
>
> Thank you for pointing out this misnomer, the caption has been adjusted accordingly. This experiment is meant to compare with implicit representation networks which are trained on a single image. We show that GONs can represent whole datasets to a high degree of fidelity. To answer your question explicitly, the images in Figure 4 are indeed all from the same trained GON.
>
> (Continued Below)

---

> > ### Author Response · Authors · 2020-11-20
> > **Response to Reviewer 2**
> >
> > > “Is Figure 7 from an explicit GON or an implicit GON? If its explicit, how are the number of parameters comparable to an implicitGON? Clearly an explicit model will have a lot more number of parameters. esp as the size of the images increase?”
> >
> > Figure 7 is from an explicit GON, the caption has been updated to clarify this. The number of parameters used for implicit GONs and explicit GONs are shown in the captions of Figure 4 and 7 respectively. While not a direct comparison since Figure 7 is a variational GON and the numbers of parameters are chosen to be approximately comparable, this does provide some insight. We find that implicit models are able to better represent data when fewer parameters are available or when the image size is larger.
> >
> > > “I really like and appreciate the variationalGON experiments. How do they compare with standard VAEs? Can they recover CelebA 64x64 images? How would they compare on quantitative metrics like FID etc.?”
> >
> > In the latest update we have included quantitative comparisons between variational GONs and VAEs in terms of ELBO on the test set in Table 2. This includes evaluation on the CelebA 64x64 dataset. In 5 of the 6 datasets tested on, the GON approach achieves lower ELBO than the VAE. Samples from the variational GON trained on CelebA can also now be found in Figure 9b in order to assess this qualitatively.
> >
> > > “In the super resolution experiment, can it super resolve any image from the distribution it was trained on? For e.g. in figure 5. is it just a matter of resampling the grid to 256x256 and running them through the pre-trained model for any sample from p(x)?”
> >
> > Yes, it can super resolve any image thanks to the generalisation ability of GONs, and it is as simple as you state; Figure 17 has been updated to contain super-samples of images in the MNIST test set.

---

### Official Review · AnonReviewer3 · 2020-10-30
**Lack of solid formulation and strong experiments**

**Rating:** 5
**Confidence:** 4

**Review:**

This paper proposes a new type of generative models with a new inference method of latent variables. Specifically, the gradient of latent variables with respect to zero vector is taken as the inferred latent variables. Based on this, the authors generalize the propose model to implicit and variational versions and demonstrate the models on image datasets.

Pros: the proposed method is easy and straightforward to implement.

Cons:
1. The model assumption that the one step gradient from zero vector equals to latent vector is quite limited and greatly constrains the model expressiveness. A justification that such assumption is reasonable is badly needed.

2. Formulation needs to be carefully checked. For example, Eqn 2 is not entirely correct to me. The second term should not be binary cross entropy as there is no categorical variable involved. Also, please avoid using abbreviations (L^BCE, L^CCE) at the first time to introduce them, which are confusing.

3. Experimental results are not sufficient to demonstrate the efficacy. Need more quantitative analysis and experiments on more challenging datasets.

4. The claim that it saves parameters compared to VAE is confusing. In the variational version, parametrizations of mu(x) and sigma(x) are also required. A principled way to very this claim is to show that with the variational version, the method could use much less parameters compared VAE while has the better synthesis quality.

Overall, the method proposed in this paper is new and promising. However, given the current unclear formulation and lack of strong experimental results, I recommend a rejection.

---

> ### Author Response · Authors · 2020-11-20
> **Response to Reviewer 3**
>
> Thank you for your very helpful feedback, we will try to address your comments point-by-point:
>
> > “The model assumption that the one step gradient from zero vector equals to latent vector is quite limited and greatly constrains the model expressiveness. A justification that such assumption is reasonable is badly needed.”
>
> We have updated the paper, deriving our approach from empirical Bayes. In summary, for a latent variable z and data point x, if we have a noisy observation of z, i.e. z_0=z+N(0,I), then empirical Bayes’ allows us to obtain the expected value of p(z|x) using a single gradient step. When a normally distributed prior is assumed over z, then we can choose z_0 as the origin since it is the mean of p(z_0).
>
> We also provide an explanation, from a function approximating perspective:
> 1. The gradient itself is a non-linear function that can approximate functions: the derivative of a deep MLP corresponds to a product of networks.
> 2. Using the gradient as an encoder offers good initialisation since it inherently provides an improved estimate of the latent vector.
> 3. Good latent spaces should satisfy local consistency (points close in latent space should be close in output space). Similar data points have similar gradients so this is satisfied. The exact gradient is thus relatively unimportant; the network’s prior must be the gradient operation but since the gradient is relatively unimportant, this does not severely restrict the network’s expressiveness.
>
> > “Formulation needs to be carefully checked. For example, Eqn 2 is not entirely correct to me. The second term should not be binary cross entropy as there is no categorical variable involved. Also, please avoid using abbreviations (L^BCE, L^CCE) at the first time to introduce them, which are confusing.”
>
> Thank you for pointing this out. The variational GON formulation has been altered to be more general.
>
> > “Experimental results are not sufficient to demonstrate the efficacy. Need more quantitative analysis and experiments on more challenging datasets.”
>
> We have added a number of additional experiments analysing the GON formulation including the effect of multiple gradient descent steps, confirming our hypothesis that a single step is sufficient (Figure 2b) and the ability for GONs to generalise (Figure 2c); qualitative experiments on more challenging datasets: reconstructions on LSUN Bedrooms (Figure 9a) and samples from a variational GON trained on CelebA (Figure 9b); and quantitative analysis comparing GONs with other approaches as suggested by Reviewer 1 where we find GONs to be competitive with multi-step approaches (Table 1) as well as a comparison between variational GONs and VAEs in terms of validation ELBO (Table 2). In a future update, more experiments will be added.
>
> > “The claim that it saves parameters compared to VAE is confusing. In the variational version, parametrizations of mu(x) and sigma(x) are also required. A principled way to very this claim is to show that with the variational version, the method could use much less parameters compared VAE while has the better synthesis quality.”
>
> To implement a variational GON we integrate the reparameterization trick into the decoder network. Specifically, the forward pass takes input z, is mapped by two linear layers to mu(z) and sigma(z), the reparameterization trick is applied, then the rest of the function is performed to obtain p(x|z). This allows us to use the GON update step to obtain z from the original z_0, while still parameterising mu and sigma. The parameters are thus reused in the derivative as the encoder so that there are just under half as many. We have attempted to clarify this in the variational GON section of the method. As suggested, we quantitatively verify this in Table 2 and find that the variational GON achieves lower validation ELBO than an equivalent VAE with almost twice as many parameters on 5 of the 6 datasets.

---

### Author Response · Authors · 2020-11-24
**Summary of revision changes**

We thank the reviewers for their valuable comments. During this rebuttal period we have made a very significant revision to the original paper, including the addition of Section 3.1, a new derivation from empirical Bayes (proof of conditional empirical bayes in Appendix A), added Table 1, showing quantitatively that GONs significantly outperform competing single-step methods, Table 2, demonstrating that variational GONs achieve substantially lower ELBO than VAEs (on 5/6 datasets) including large complex datasets (CelebA), Figure 9, showing qualitatively that GONs can well represent large complex datasets (added higher resolution LSUN Bedrooms and CelebA), Figure 2b, confirming that a single gradient step is sufficient, Figures 2c and 16, demonstrating GON generalisation, Figure 17, showing that GONs allow superresolution on test data, Table 1 and Figures 2a, 3, and 10 have been updated to include means/standard deviations (error envelopes) from multiple runs, alongside various new discussions.

---

### Comment · ~Soochan_Lee1 · 2021-03-14
**GON is a linear model**

I think I found a critical problem with GON.

The latent variable $z$ is computed as follows:
$$
\begin{align}
z
&= -\frac{\partial \mathcal L^{MSE}}{\partial z} \bigg\vert_{z=0} \\\\
&= - \frac{\partial \mathcal L^{MSE}}{\partial F}\bigg\vert_{z=0} \frac{\partial F}{\partial z}\bigg\vert_{z=0} \\\\
&= - (x - F(\mathbf 0)) \underbrace{\frac{\partial F}{\partial z} \bigg\vert_{z=0}}_{\text{const. w.r.t. } x}
\end{align}
$$
Here we can see that $z$ is just an affine transformation of $x$.
There is no nonlinearity.
Therefore, we can conclude that GON is practically equivalent to a linear model.

Here is [a Colab notebook](https://colab.research.google.com/drive/1xIY5CEUFilASnuonWChTiABAouRu5Bup?usp=sharing) that shows GON's limitation with a simple toy example.
Note that I extended the Colab notebook provided by the authors.
I created a set of random points on the surface of the 3D unit sphere, which is a 2D manifold.
An MLP autoencoder can successfully reconstruct this data distribution with 2D latent variables, while GON cannot.

The same is true for variational GON. Adding one more linear layer to extract variational parameters does not add any nonlinearity.

I was initially very excited by the authors' claim that we can learn a deep generative model without an encoder, but I am afraid this is not true.
The high-quality examples in the experiment section would be due to a large $z$ dimension.

---

> ### Author Response · Authors · 2021-03-15
> **GONs are not linear models**
>
> There's an error in your reasoning, we define the latent variable computation in terms of $p(x|z)$ (Equation 7) which can be modelled such that the components of $x$ have conditional dependencies, causing $\frac{\partial F}{\partial z}$ to be dependent on $x$. A simple counter example to your unit sphere example is to model $p(x|z)$ using a conditional MADE [1], which can easily reconstruct the data distribution with 2D latent variables, without an encoder. Here is a [Colab notebook](https://colab.research.google.com/gist/samb-t/d3a4d7d7204bda3a0c81995e654d00d4/made-gon-sphere.ipynb) demonstrating this.
>
> As for the high-quality samples being just due to a large $z$ dimension, please see Figure 10c where we demonstrate GONs reconstruction ability for a variety of latent sizes. Indeed, GONs outperform autoencoders even with small latent spaces.
>
> [1] Germain, M., Gregor, K., Murray, I., & Larochelle, H. (2015). MADE: Masked Autoencoder for Distribution Estimation. In International Conference on Machine Learning (pp. 881-889). PMLR.

---

> > ### Comment · ~Soochan_Lee1 · 2021-03-15
> > **There is no error in my reasoning**
> >
> > Yes, GONs compute gradient w.r.t. $\log p(x|z)$.
> > This is equivalent to computing gradient w.r.t. MSE loss if $\log p(x|z)$ is assumed to be a Gaussian with unit covariance.
> > This is also explicitly expressed in Eq. (8).
> >
> > My point is that the GON's encoding capability is severely limited (i.e., linear encoding when used with the MSE loss).
> > I clearly demonstrated this fact with the notebook that I shared earlier.
> > If I am wrong, please point out which part of my code is incorrect.
> > According to my understanding, every experiment in the paper shares this limitation.
> >
> > My code shows that GONs cannot model a simple 2D manifold with 2D latent variables, while autoencoders can.
> > The authors' response is completely irrelevant to this issue.
> > They argue that MADE, an acronym for Masked ***Autoencoder*** for Distribution Estimation, can be used to model the 2D manifold.
> > But this is utterly obvious because MADE has a nonlinear encoder!
> > The main novelty claimed by this paper is that GONs can infer latent variables without an encoder.
> > I cannot understand why the authors brought this up as a counterexample.

---

> > > ### Author Response · Authors · 2021-03-15
> > > **GONs are not mean-field models and this is not a main limitation**
> > >
> > > When the generative function models each data component independently, the encoding function is linear, however, this is not a fundamental property of GONs since conditional dependencies can be modelled effectively. We'll make sure we make no such claims in the final camera ready version. Besides, GONs seem to excel in high dimensional data spaces as demonstrated by our strong empirical results, significantly outperforming autoencoders in real world cases, fitting better while also generalising better, even with small latent vectors despite making this simple independence assumption. This allows applications such as modelling $p(x|z)$ using an implicit network with advantages such as superesolution.
> > >
> > > Additionally, MADE is not a standard autoencoder, it is an autoregressive model obtained by masking an autoencoder’s weights such that each output neuron is conditioned only on the previous values. This is different to a standard autoencoder which, unlike MADE, models the components of x independently but requires a bottleneck to prevent the identity function from being learned. Our example demonstrates this in a latent conditional model, comparable to a PixelVAE [1] or MAE [2], both of which require an encoder to compress data to latent vectors. On the other hand, our example demonstrates that GONs can achieve this without an encoder.
> > >
> > > [1] Gulrajani, I., Kumar, K., Ahmed, F., Taiga, A. A., Visin, F., Vazquez, D., & Courville, A. (2016). Pixelvae: A latent variable model for natural images. ICLR.
> > >
> > > [2] Ma, X., Zhou, C., & Hovy, E. (2019). MAE: Mutual posterior-divergence regularization for variational autoencoders. ICLR.

---

> > > > ### Comment · ~Soochan_Lee1 · 2021-03-16
> > > > **The authors are obscuring the point**
> > > >
> > > > Unfortunately, the authors continue to make inaccurate and irrelevant claims.
> > > > Here I summarize several key facts:
> > > >
> > > > 1. [This notebook](https://colab.research.google.com/drive/1EhBdvsuNHRhAtOidYu59JTyzZXfrUpxN?usp=sharing) shows that a linear encoding achieves the same reconstruction loss as GON encoding in MNIST and Fashion MNIST. It is unarguably clear, in both theoretical and empirical aspects, that GON encodes linearly.
> > > >
> > > > 1. MADE has an encoder part. To be specific, the first component in their code (`MaskedLinear(nin, nhidden, num_cond_inputs, masks[0])`) is the encoder. Meanwhile, the first sentence in the abstract of this paper is, "This paper proposes a new type of generative model that is able to quickly learn a latent representation **without an encoder**."
> > > >
> > > > 1. The authors did not even cite MADE in their paper.
> > > >
> > > > 1. Using an autoregressive decoder is out of the scope of this paper. The authors did not mention nor experiment with the idea.
> > > >
> > > > 1. The authors did not show how to combine GON with an autoregressive model without an encoder. I doubt that using a series of linear models to autoregressively decode would add any nonlinearity.
> > > >
> > > >
> > > > I feel very sorry for the authors, but I have to suggest withdrawing this paper.
> > > > It would cause a lot of confusion and waste many valuable hours of other researchers.
> > > > I hope the authors make a wise decision for the community.

---

> > > > > ### Author Response · Authors · 2021-03-16
> > > > > **We are not obscuring the point**
> > > > >
> > > > > 1. Using a linear encoding like this is clearly not scalable. To show our point, for our bedrooms experiment (128x128x3 compressed to 2048), your above notebook example with a linear encoder would use **100,663,296 parameters** whereas **GONs use 0**. And the qualitative results are impressive in both cases.
> > > > > 	- GONs are presented in a general form; we don't dispute that mean-field GONs encode linearly. Our high dimensional test cases show they outperform their non-linear AE equivalents (which are standard convolutional AE architectures).
> > > > > 	- As stated throughout our abstract and paper, one of the main use cases of GONs is their applications in implicit networks.
> > > > > 2. As discussed in our previous comment, MADE is an autoregressive model.
> > > > > 3. We just used MADE as a simple example to demonstrate GONs capability for your toy benchmark. Other likelihood models such as PixelVAE or MAE can be used for higher dimensional data.
> > > > > 4. In the method section of our paper we discuss modelling in broad terms of $p(x|z)$ and do not state that this has to be a mean-field model. Our experiments demonstrate the efficacy of GONs with simple independent output distributions outperforming autoencoders while using significantly fewer parameters, but GONs can also be applied to more complex decoders.
> > > > > 5. Consider an autoregressive model $p(x_1|z)p(x_2|x_1,z)\cdots p(x_n|x_{<n},z)$. The gradient of $p(x_2|x_1,z)$ with respect to $z$ is dependent on $x_1$ and the gradient of $p(x_n|x_{<n},z)$ with respect to $z$ is dependent on $x_{<n}$. This is what the code we shared in our initial answer does.
> > > > >
> > > > > To summarise our position:
> > > > > - GONs are presented in a general form that supports both linear and non-linear encodings dependent on how $p(x|z)$ is modelled.
> > > > > - Linearly encoding GONs significantly outperform autoencoders on high dimensional data with substantially fewer parameters, while converging faster and generalising better.
> > > > >
> > > > > In light of your strong views on this, we are more than happy to add a statement to our paper that the mean-field implementation of GONs equates to a linear encoding and recheck, to make sure we are consistent with such claims - this does not otherwise affect the narrative, contributions, or any part of this paper: abstract, introduction, method, results, discussion, and conclusions.

---

> ### Author Response · Authors · 2021-03-18
> **Multi-step GONs are non-linear**
>
> We agree that single-step GONs for mean-field implementations are linear as you've helpfully shown clearly above. However, multiple-step GONs (see Figure 2b and Table 1) for this are able to give non-linear encodings (because the second step uses a $z$ that is a linear function of $x$ so now $F(z)$ is dependent on $x$). In our experiments we found that the non-linearity induced by multiple-steps didn’t provide quantitative improvement when dealing with high-dimensional datasets while it significantly increased run-time (see Figure 2b and Table 1).
>
> Here is a [Colab notebook](https://colab.research.google.com/gist/cwkx/1f3db3c088334fdccb24822ee280bb2a/non-linear-2-step-gon-encodings-example.ipynb) demonstrating a 2-step GON giving clear non-linear encodings on your sphere test example.

---

### Decision · Program_Chairs · 2021-01-07
**Final Decision**

**Decision:**

Accept (Poster)

**Comment:**

This paper presents a new inference mechanism for latent variable models, by taking the derivative of log-likelihood with respect to a zero-valued vector. Initially, the reviewers raised concerns mostly regarding the limited experimentation and missing baselines. However, in the revised version, the authors addressed most of these concerns.

Given that most reviewers are positive after the revision and since the proposed method is simple and interesting, I recommend accepting this paper.